# MolReasoner: Toward Effective and Interpretable Reasoning for Molecular LLMs

## Abstract

Large Language Models (LLMs) have demonstrated remarkable performance across various domains, yet their capabilities in molecular reasoning remain insufficiently explored. Current approaches tend to rely heavily on general-purpose prompting, which lacks domain-specific molecular semantics, while those that use fine-tuning strategies often face challenges with interpretability and reasoning depth. To address these issues, we introduce **MolReasoner**, a two-stage framework designed to transition LLMs from memorization towards chemical reasoning. First, we propose Mol-SFT, which initializes the model's reasoning abilities by distilling high-quality reasoning Chain-of-Thought (CoT) trajectories from GPT-4o, enriched with structural features and functional group information, and verified for chemical accuracy, enabling the model to internalize coherent and chemically meaningful reasoning. In the Mol-RL stage, we propose verifiable and extensible multi-level rewards, where language- and structural-similarity rewards provide fine-grained semantic and structural alignment. Moreover, we introduce more comprehensive metrics, together with a multi-dimensional expert-aligned pipeline to rigorously assess reasoning quality. Extensive experiments demonstrate that MolReasoner outperforms existing methods, and marking a significant shift from memorization-based outputs to robust chemical reasoning. The code for MolReasoner is included in the supplementary materials and will be open-sourced in the near future.

## 1 Introduction

Given the significance of molecular science (Moore et al., 2002) — spanning applications such as drug discovery (Drews, 2000) and materials design (Tibbitt et al., 2015), — as well as the growing need for deeper insights into molecular structures, molecular reasoning tasks have become vital for uncovering underlying chemical relationships and intrinsic patterns in molecular data, directly supporting molecular design, property prediction, and the generation of novel molecules. In parallel, large language models (LLMs) (Hurst et al., 2024; Team et al., 2023; Yang et al., 2025) have recently achieved impressive capabilities across diverse domains, prompting interest in utilizing LLMs to enhance molecular comprehension and generation. To bridge molecular science and language modeling, one notable research direction is the translation between molecular descriptors and natural language (Edwards et al., 2022).

The extensive knowledge and reasoning capabilities of LLMs make them promising candidates for complex text-related molecular tasks. Early **Prompt-based approaches** (Wu et al., 2024; Guo et al., 2023; Liu et al., 2024b) utilize general-purpose LLMs without domain-specific fine-tuning, relying on intentionally engineered instructions or in-context examples. However, as illustrated in Figure 1, these methods are limited in capturing molecular semantics due to the absence of chemistry-specific adaptation, leading to a heavy reliance on superficial linguistic cues or templates rather than accurately encoding underlying chemical structures. This issue can result in detrimental errors, including incorrect atom counts and the generation of chemically implausible structures, underscoring the limitations of general-purpose LLMs in effectively reasoning about molecular content.

Subsequently, **Fine-tuning methods without explicit reasoning** (Fang et al., 2024; Yu et al., 2024; Liu et al., 2023; Li et al., 2024) emerged, training LLMs directly on molecule-text pairs formatted as question-answer or instruction-following tasks. However, as shown in Figure 1, these methods

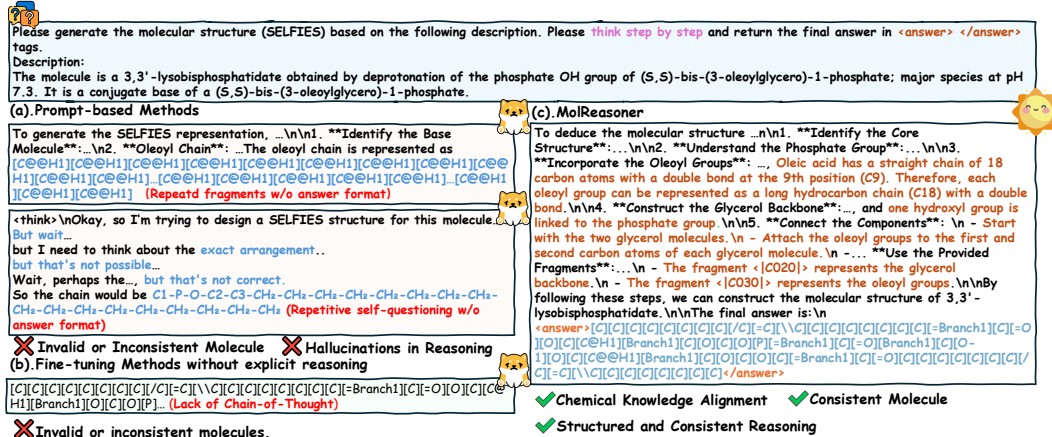

Figure 1: Examples of text-based molecule generation. (a) **Prompt-based methods** often hallucinate and yield chemically invalid molecules due to a lack of chemistry-specific adaptation. (b) **Fine-tuning without explicit reasoning** encourages memorization over generalization, reducing interpretability. (c) **MolReasoner** provides structure-grounded Chain-of-Thought reasoning, yielding interpretable and chemically valid candidates.

typically utilize supervision signals limited exclusively to final outputs, lacking explicit guidance through intermediate reasoning steps, causing supervised memorization rather than genuine molecular reasoning. Therefore, the models are incapable of genuinely internalizing chemical principles and logical reasoning processes, severely limiting their generalization ability toward unseen molecular structures (Chu et al., 2025). Furthermore, the absence of explicit reasoning undermines interpretability, which in turn reduces model reliability and restricts its applicability in high-stakes molecular applications. These limitations underscore that existing methods fail to fully exploit the potential of LLMs in molecular tasks, largely due to insufficient reasoning capabilities. Motivated by the above observation, we pose and try to answer the central research question: *Can we go beyond mere memorization and teach LLMs to reason on molecular tasks?*

A natural approach to addressing this issue is to construct Chain-of-Thought (CoT) data (Wei et al., 2022), which provides explicit reasoning processes alongside question-answer pairs. However, manually crafting such detailed reasoning is prohibitively costly, as it demands substantial input from domain experts. Recent advances in Large Reasoning Models (LRMs) (Jaech et al., 2024; Liu et al., 2024a) propose an alternative by leveraging Reinforcement Learning (RL) to promote reasoning based solely on outcome-based supervision. While this reduces the reliance on handcrafted reasoning data, it encounters another obstacle — the *cold-start* problem. Without prior reasoning guidance or domain-specific adaptation, models initially struggle to discover effective reasoning strategies, resulting in sparse reward signals early in training.

To address these challenges and shift molecular LLMs from memorization toward genuine reasoning, we propose **MolReasoner**, a two-stage training framework comprising **Mol-SFT** for initializing reasoning procedures and **Mol-RL** for activating deeper internal chemical reasoning knowledge. Specifically, Mol-SFT leverages molecule–text pairs augmented with reasoning trajectories generated via GPT-4o (Hurst et al., 2024), further enriched with structural features and functional group information to ensure coherent and chemically meaningful reasoning. These distilled trajectories allow the model to internalize reasoning formats and acquire initial molecular reasoning capability. Since format-only rewards are inadequate for complex chemical knowledge, Mol-RL enhances reasoning with verifiable multi-level rewards, where language- and structural-similarity signals provide fine-grained semantic and structural alignment. Ablation studies show that both rewards are effective and indispensable, providing an extensible paradigm for molecular reinforcement learning.

Besides, we introduce fragment- and functional group–level metrics to better assess structural consistency and hallucination, and design a multi-dimensional expert-aligned evaluation. Results and case studies show that MolReasoner produces significantly higher-quality trajectories than existing baselines, validating its advantage in molecular reasoning. Overall, MolReasoner represents a paradigm shift from memorization to genuine reasoning: instead of merely recalling training an-

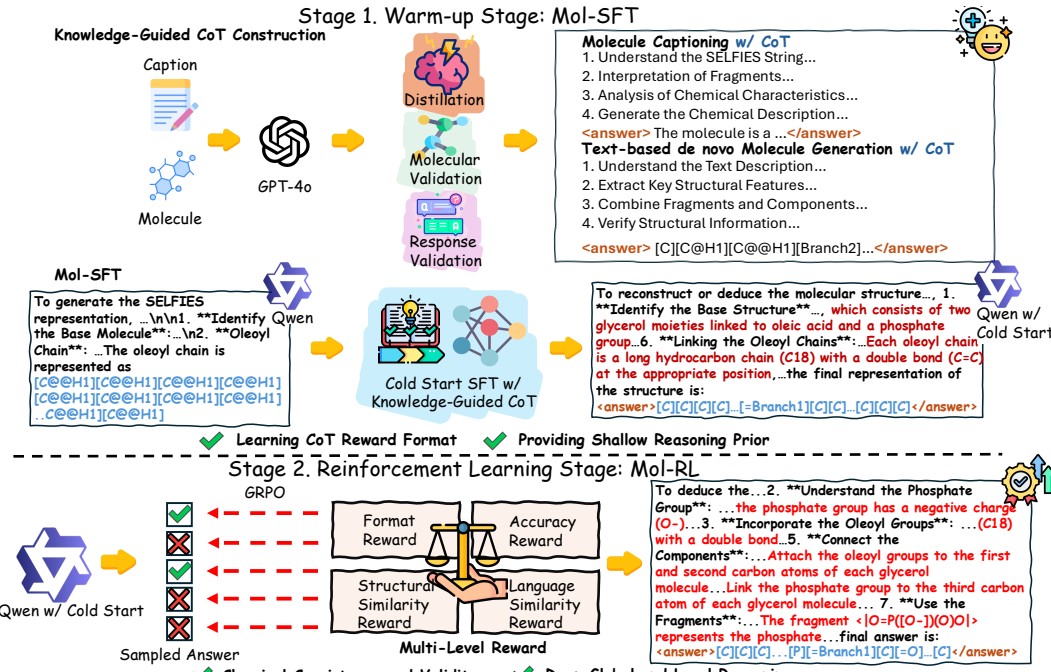

Figure 2: MolReasoner is a two-stage training framework: (1) Mol-SFT initially utilizes molecule–text pairs, augmented by reasoning trajectories generated via GPT-4o, to bootstrap reasoning capabilities; and (2) Mol-RL subsequently refines the reasoning ability through a carefully designed reward function that encourages precise alignment between molecular structures and their corresponding textual descriptions.

swers, the model learns to understand molecular structures and their associated textual semantics, generating solutions grounded in coherent and interpretable reasoning.

## 2 METHODOLOGY

In the course of this work, we dedicate to the molecule-text translation task (Edwards et al., 2022), which evaluates a model's capability to align molecular representations with corresponding natural language descriptions through effective reasoning. Further details can be found in the appendix. Additionally, the appendix includes a comprehensive review of related research on molecular language models and large-scale reasoning models. In the following section, we present the training pipeline of MolReasoner. Started with detailing the construction of CoT data for cold-start, we then introduce how the Mol-SFT and Mol-RL stages enhance the molecular reasoning abilities of the model.

### 2.1 KNOWLEDGE-GUIDED CoT DATA CONSTRUCTION

To warm up before RL, we propose a knowledge-guided CoT data construction pipeline tailored specifically for molecular tasks, which aims to familiarize LLMs with interpretable reasoning processes. There are two molecular translation tasks: 1) molecule captioning and 2) text-based de novo molecule generation. Leveraging the ChEBI-20 (Edwards et al., 2021) training set, which provides paired records of molecules and their natural-language descriptions, we construct CoT datasets using tailored prompt template, chemical knowledge injection, and rigorous output filtering strategies.

Inspired by prior work (Jang et al., 2024b; Ouyang et al., 2024; Jang et al., 2024a) incorporating structural knowledge to promote model reasoning, we further enrich our prompts with comprehensive and detailed chemical structure information. As showcased in the appendix, we extract statistical structural features (e.g., , number of rings, aromaticity, molecular weight) to anchor the model's understanding of molecular structures. Additionally, considering the crucial role of functional groups in defining molecular properties, we adopt the EFGs (Lu et al., 2021) approach to iden-

tify key functional groups and hydrocarbon fragments, integrating these into prompts to strengthen the alignment between chemical structures and linguistic descriptions. This multi-level guidance — combining structural descriptors and functional fragments — enables the model to produce coherent and chemically meaningful reasoning trajectories.

While previous studies (Taylor et al., 2022; Edwards et al., 2022) use SMILES for molecule representation, its grammar and token order sensitivity often result in invalid or chemically implausible outputs. To overcome this limitation, Mol-Instructions (Fang et al., 2024) adopts the SELFIES (Krenn et al., 2022) representation. SELFIES ensures chemical validity by construction, eliminating common structural errors such as mismatched parentheses, invalid atomic symbols, and illogical branching patterns.

With these foundations, for the **molecule captioning** task, given an input molecule represented as a SELFIES sequence, we employ a unified prompt template guiding GPT-4o (Hurst et al., 2024) to reason step-by-step using the provided structural information. Generated reasoning trajectories are then distilled into semantically accurate captions. Through stringent semantic consistency filtering, we obtain approximately 18,000 high-quality CoT samples. For the more challenging **text-based de novo molecule generation** task, prompts are designed to simulate the reasoning process of chemists, explicitly guiding the model to identify molecular components from textual descriptions and systematically assemble them into valid SELFIES. After filtering for structural validity and semantic coherence, we acquire approximately 24,000 high-quality CoT samples. Notably, although both tasks originate from the same molecular dataset, differences in task directionality, CoT prompt design, and filtering criteria result in varying numbers of high-quality samples.

## 2.2 WARM-UP STAGE: MOL-SFT

The effectiveness of LLMs in molecular tasks is often constrained by poor structure–semantic alignment, incoherent reasoning trajectories, and limited comprehension of chemical semantics. Thus, we introduce the Molecular Supervised Fine-Tuning (**Mol-SFT**) stage, designed to establish foundational reasoning formats and linguistic logic tailored specifically to molecular tasks.

In the Mol-SFT stage, we utilize synthetic CoT examples to explicitly guide the model in learning step-by-step reasoning processes. Specifically, we employ approximately 42,000 high-quality CoT samples from our constructed molecule captioning and text-based generation datasets. Training is performed via a standard autoregressive language modeling objective:

$$\mathcal{L}_{\text{Mol-SFT}}(\theta) = -\sum_{t=1}^{T} \log p(y_t \mid x, y_{<t}; \theta) \tag{1}$$

where $x$ denotes the instruction prompt, $\{y_1, y_2, \ldots, y_T\}$ represents the reasoning trajectory and final answer, and $\theta$ denotes the model parameters. This stage enables the model to internalize molecular reasoning formats, domain-specific terminology, and linguistic structures via example-based learning. After Mol-SFT, the model demonstrates an enhanced capability to follow structured reasoning instructions and generate coherent reasoning chains. This warm-up phase provides shallow reasoning priors and format awareness, laying essential groundwork for subsequent reinforcement learning-based refinement.

## 2.3 REINFORCEMENT LEARNING STAGE: MOL-RL

To further refine molecular reasoning capabilities, we introduce **Mol-RL**, which guides the model in optimizing both reasoning pathways and generation outcomes. In this stage, we employ Group Relative Policy Optimization (GRPO), where the policy model $\pi_\theta$ generates multiple candidate responses for each input. These candidate responses are then evaluated using tailored, chemically verifiable reward functions, which provide explicit feedback signals to encourage high-quality and chemically coherent reasoning trajectories. The GRPO algorithm is detailed in the appendix. For molecule captioning, we introduce a verifiable reward function $R_{\text{cap}}$ designed to encourage correct formatting and semantically accurate natural-language descriptions:

$$R_{\text{cap}} = \begin{cases} 0.5 + 1.5 \times R_{\text{language}}, & \text{if format is correct,} \\ 0.0, & \text{if format is incorrect.} \end{cases} \tag{2}$$

The format is considered correct if the final output is enclosed within `<answer>...</answer>` tags. The language-similarity reward $R_{\text{language}}$ is computed as the mean of six standard evaluation metrics (BLEU-2, BLEU-4, METEOR, ROUGE-1, ROUGE-2, and ROUGE-L) (Papineni et al., 2002; Banerjee & Lavie, 2005; Lin, 2004) comparing the generated caption to the reference description.

In the text-based de novo molecule generation task, we propose a verifiable reward $R_{\text{gen}}$ to improve the structural integrity and semantic alignment of generated molecules:

$$R_{\text{gen}} = \begin{cases} 0.5 + 1.5 \times R_{\text{structural}}, & \text{if format is correct,} \\ 0.0, & \text{if format is incorrect.} \end{cases} \tag{3}$$

where the structural-similarity reward $R_{\text{structural}}$ is computed as the mean of four key components — fingerprint similarity, SELFIES-level language similarity, fragment similarity, and functional group matching — to provide a holistic and chemically aware measure of structural alignment.

$$R_{\text{structural}} = \frac{1}{4} \left( \text{FP}_{\text{sim}} + \text{SELFIES}_{\text{sim}} + \text{FRAG}_{\text{sim}} + \text{FG}_{\text{match}} \right) \tag{4}$$

We compute fingerprint similarity based on three widely used molecular fingerprints: Morgan, MACCS, and RDKit (Tanimoto, 1958; Durant et al., 2002). The final score is calculated as the mean of these three, providing a balanced measure of molecular similarity:

$$\text{FP}_{\text{sim}} = \frac{1}{3} \left( \text{Morgan} + \text{MACCS} + \text{RDKit} \right) \tag{5}$$

Additionally, $\text{SELFIES}_{\text{sim}}$ is assessed by computing the character-level BLEU score between the predicted and reference SELFIES sequences.

In addition to the two similarity-based rewards, we observe that fragment and functional group hallucinations are regularly presented during the molecular reasoning process. Even though the generated molecules may be chemically valid, they can exhibit significant inconsistencies with the reference structures. To address this issue, we introduce two additional rewards: fragment similarity and functional group matching. We use EFGs (Lu et al., 2021) to identify differing fragments between the generated molecules and the reference molecules. Fragment similarity is then computed by jointly considering the Jaccard overlap and fragment-level recall, effectively capturing both coverage and precision of structural subcomponents. Here, $\mathcal{F}_{\text{pred}}$ and $\mathcal{F}_{\text{ref}}$ refer to the sets of fragments in the predicted and reference molecules, respectively. The final fragment similarity score is given by:

$$\text{FRAG}_{\text{sim}} = 0.5 \times \frac{|\mathcal{F}\text{pred} \cap \mathcal{F}\text{ref}|}{|\mathcal{F}\text{pred} \cup \mathcal{F}\text{ref}|} + 0.5 \times \frac{|\mathcal{F}\text{pred} \cap \mathcal{F}\text{ref}|}{|\mathcal{F}\text{ref}|} \tag{6}$$

Additionally, the functional group matching reward computes the difference in the number of functional groups, excluding CH-only groups, between the predicted and reference molecules, using an exponential decay formulation.

$$\text{FG}_{\text{match}} = \exp\left( -\frac{\sum_k |\text{count}_{\text{pred}}(k) - \text{count}_{\text{ref}}(k)|}{\sum_k \text{count}_{\text{ref}}(k) + \varepsilon} \right) \tag{7}$$

where $\text{count}_{\text{pred}}(k)$ refers to the number of occurrences of functional group $k$ in the predicted molecule, and $\text{count}_{\text{ref}}(k)$ refers to that of functional group $k$ in the reference molecule. We set $\varepsilon = 10^{-5}$ to ensure numerical stability.

By incorporating chemical awareness and granularity through multi-level reward feedback, the model learns to align chemical knowledge from global molecular semantics to local molecular structural details during inference, thereby ensuring greater consistency in generated chemical structure. As a result, the model transitions from merely producing "valid" molecules to generating "high-quality" structures that are both chemically coherent and semantically aligned with the input, improving its generation capabilities.

## 3 EXPERIMENTS

This section outlines our experimental setup and presents the results that demonstrate the effectiveness of **MolReasoner**. Due to the space limits, we refer to the appendix for implementation details.

| Method | Size | BLEU-2↑ | BLEU-4↑ | METEOR↑ | ROUGE-1↑ | ROUGE-2↑ | ROUGE-L↑ |
|---|---|---|---|---|---|---|---|
| *Closed-Source Model* | | | | | | | |
| GPT-4o | - | 0.1194 | 0.0433 | 0.1651 | 0.2315 | 0.0738 | 0.1792 |
| GPT-4o-mini | - | 0.1080 | 0.0400 | 0.1545 | 0.2310 | 0.0723 | 0.1776 |
| *Open-Source Model* | | | | | | | |
| Qwen2.5-7B-Instruct | 7B | 0.0792 | 0.0258 | 0.2132 | 0.2091 | 0.0601 | 0.1483 |
| DeepSeek-R1-Qwen-7B | 7B | 0.1173 | 0.0469 | 0.1544 | 0.2209 | 0.0749 | 0.1693 |
| LLaMA3.1-8B-Instruct | 8B | 0.1670 | 0.0769 | 0.2164 | 0.2806 | 0.1182 | 0.2250 |
| Qwen3-8B | 8B | 0.0974 | 0.0289 | 0.1733 | 0.2067 | 0.0501 | 0.1567 |
| LLaMA3.1-70B-Instruct | 70B | 0.1466 | 0.0658 | 0.1832 | 0.2736 | 0.1072 | 0.2203 |
| Qwen2.5-72B-Instruct | 72B | 0.1519 | 0.0647 | 0.1949 | 0.2729 | 0.0948 | 0.2067 |
| LLaMA3 + MSR | 8B | 0.0993 | 0.0237 | 0.1657 | 0.2274 | 0.0595 | 0.1559 |
| Mol-Instructions + LLaMA2 | 7B | 0.0956 | 0.0667 | 0.1891 | 0.2801 | 0.1823 | 0.2582 |
| Mol-Instructions + LLaMA3 | 8B | 0.0060 | 0.0023 | 0.0168 | 0.0124 | 0.0023 | 0.0119 |
| **MolReasoner (Ours)** | 7B | **0.4383** | **0.3220** | **0.4754** | **0.5530** | **0.3662** | **0.4821** |

Table 1: Performance of Molecule Captioning. MolReasoner outperforms all closed-source and open-source baselines across BLEU-2/4, METEOR, and ROUGE metrics, establishing the new state-of-the-art in this molecule captioning task.

## 3.1 DATASETS

With validity-based filtering criteria, we generate approximately 42,000 high-quality CoT samples from the training set of ChEBI-20 — 24,000 samples for molecule captioning and 18,000 for text-based de novo molecule generation. These samples form for the initial warm-up training stage. For the later reinforcement learning stage, we construct two GRPO training datasets derived from ChEBI-20.

## 3.2 EVALUATION AND BASELINES

We utilize the test set of ChEBI-20 as our benchmark. The molecules are represented using SELF-IES, following Mol-Instructions (Fang et al., 2024). Evaluation is performed using task-specific metrics.

**Molecule Captioning.** Following Mol-Instructions (Fang et al., 2024), we adopt standard language generation metrics, including BLEU, ROUGE, and METEOR, to assess the similarity between generated molecular descriptions and ground-truth references.

**Text-based de novo Molecule Generation.** We use RDKit to validate the chemical correctness of generated strings and compute exact match rates. In addition, we evaluate molecular similarity using Tanimoto scores, Levenshtein distance, and BLEU scores.

To further assess the structural fidelity of generated molecules, we propose three fragment-level metrics: (1) **Frag-J:** Measures the Jaccard similarity between the fragment sets of the predicted and reference molecules. Higher values indicate larger structural overlap. (2) **Frag-R:** Assesses whether key structural fragments in the reference molecule are recalled in the prediction. (3) **FG-Match (Functional Group Matching):** Computes the difference in the number of functional groups between the prediction and the reference using an exponential decay formulation; a higher score indicates greater similarity. Unlike Mol-Instructions, which evaluates only valid molecules, we report all metrics across the entire set of generated molecules to capture overall model performance.

**Baselines.** Our primary goal is to examine how general-purpose LLMs can be adapted for molecular reasoning when equipped with explicit reasoning strategies. Therefore, we deliberately select baselines that are LLM-based models. For prompt-based methods, we compare our model against leading general-purpose LLMs, including GPT-4o, GPT-4o-mini, Qwen2.5-7B-Instruct, DeepSeek-R1-Distill-Qwen-7B, LLaMA3.1-8B-Instruct, Qwen3-8B, LLaMA3.1-70B-Instruct, Qwen2.5-72B-Instruct, and LLaMA3 with MSR (Jang et al., 2024a). In addition, we include Mol-Instructions combined with LLaMA2 and LLaMA3 as representative fine-tuning approaches that do not explicitly incorporate reasoning, thereby providing strong LLM-based molecular baselines. We use Qwen2.5-7B-Instruct as our base model for all experiments; further training details are provided in the appendix.

| Method | Size | BLEU↑ | Exact↑ | Levenshtein↓ | RDK FTS↑ | MACCS FTS↑ | MORGAN FTS↑ | Frag-J↑ | Frag-R↑ | FG-Match↑ | VALIDITY↑ |
|---|---|---|---|---|---|---|---|---|---|---|---|
| | | | | *Closed-Source Models* | | | | | | | |
| GPT-4o | - | 0.1949 | 0.0045 | 49.3545 | 0.0926 | 0.2066 | 0.0836 | 0.1296 | 0.1777 | 0.3753 | 0.2916 |
| GPT-4o-mini | - | 0.0522 | 0.0058 | 49.1371 | 0.0863 | 0.2032 | 0.0883 | 0.0987 | 0.1324 | 0.3898 | 0.1946 |
| | | | | *Open-Source Models* | | | | | | | |
| Qwen2.5-7B-Instruct | 7B | 0.0002 | 0.0024 | 40.0076 | 0.0776 | 0.1585 | 0.0520 | 0.0773 | 0.1037 | 0.3601 | 0.2395 |
| DeepSeek-R1-Qwen-7B | 7B | 0.0000 | 0.0018 | 50.6957 | 0.0619 | 0.1327 | 0.0461 | 0.1101 | 0.1428 | 0.3847 | 0.0697 |
| LLaMA3.1-8B-Instruct | 8B | 0.0094 | 0.0027 | 40.2092 | 0.0556 | 0.1470 | 0.0470 | 0.0701 | 0.0918 | 0.3587 | 0.2319 |
| Qwen3-8B | 8B | 0.0000 | 0.0036 | 28.2564 | 0.3692 | 0.4733 | 0.3059 | 0.3406 | 0.3566 | 0.5280 | 0.0118 |
| LLaMA3.1-70B-Instruct | 70B | 0.0787 | 0.0055 | 44.1626 | 0.0824 | 0.2323 | 0.0785 | 0.1398 | 0.1963 | 0.3574 | 0.4641 |
| Qwen2.5-72B-Instruct | 72B | 0.0000 | 0.0048 | **18.0588** | 0.1584 | 0.3456 | 0.1432 | 0.1696 | 0.2300 | 0.3436 | 0.1134 |
| LLaMA3 + MSR | 8B | 0.1227 | 0.0000 | 49.2581 | 0.1194 | 0.2401 | 0.0629 | 0.1690 | 0.2650 | 0.2457 | 0.9412 |
| Mol-Instructions+ LLaMA2 | 7B | 0.3049 | 0.0470 | 39.4268 | 0.2914 | 0.4427 | 0.2524 | 0.3333 | 0.4092 | 0.4324 | **0.9994** |
| Mol-Instructions+ LLaMA3 | 8B | 0.3470 | 0.0737 | 40.5716 | 0.3341 | 0.4718 | 0.2936 | 0.3868 | 0.4499 | 0.4882 | 0.9127 |
| **MolReasoner (Ours)** | 7B | **0.7841** | **0.0758** | 26.9255 | **0.4373** | **0.6759** | **0.3627** | **0.5213** | **0.6414** | **0.5390** | 0.9679 |

Table 2: Performance of Text-based de novo Molecule Generation. MolReasoner surpasses both closed-source and open-source baselines across all metrics, achieving state-of-the-art performance in this molecule generation task.

## 3.3 MAIN RESULTS

In this experiment, we compare MolReasoner with prompt-based methods and fine-tuning methods without explicit reasoning, evaluating its performance on two tasks: 1) molecule captioning (Table 1) and 2) text-based de novo molecule generation (Table 2). Across all tables, **bold** indicates the best and underline the second-best results. The results demonstrate that MolReasoner shows significant advantages in both reasoning capability and generation quality. In the molecule captioning task, MolReasoner outperforms multiple metrics. For example, the BLEU-2 score is 0.4383, which is 2.62 times higher than the best baseline; the BLEU-4 score is 0.3220, which is 4.19 times higher than the best baseline; the METEOR score is 0.4754, which is 2.20 times higher than the best baseline; the ROUGE-1 score is 0.5530, which is 1.97 times higher than the best baseline; the ROUGE-2 score is 0.3662, which is 2.01 times higher than the best baseline; and the ROUGE-L score is 0.4821, which is 1.87 times higher than the best baseline. In the text-based de novo molecule generation task, MolReasoner achieved a BLEU score of 0.7841, which is 2.57 times higher than the best baseline; an Exact score of 0.0758, which is 1.61 times higher than the best baseline; a RDK FTS score of 0.4373, which is 1.18 times higher than the best baseline; a MACCS FTS score of 0.6759, which is 1.43 times higher than the best baseline; a MORGAN FTS score of 0.3627, which is 1.19 times higher than the best baseline; a Frag-J score of 0.5213, which is 1.53 times higher; a Frag-R score of 0.6414, which is 1.80 times higher than the best baseline; and a FG-Match score of 0.5390, which is 1.02 times higher than the best baseline.

Compared to prompt-based methods without domain adaptation, MolReasoner shows notable improvements in semantic understanding and structural accuracy. These general-purpose models typically lack in-depth chemical knowledge, leading to issues such as functional group hallucinations, SELFIES being valid but semantically off-track, and ignoring structural details. By guiding the reasoning process, MolReasoner generates more accurate and chemically plausible descriptions. Models such as DeepSeek-R1-Distill-Qwen-7B, Qwen3-8B, and Qwen2.5-72B-Instruct, despite incorporating reasoning during pretraining, still suffer from conflicting reasoning chains, semantically collapsed CoT, and misleading captions, resulting in poor BLEU and validity scores; representative cases are shown in the appendix. In molecule captioning, these models often fail to capture the overall structure and chemical properties of the molecules, and the generated descriptions may contain misleading information, impacting the final molecular understanding.

Compared to fine-tuning without explicit reasoning method — Mol-Instructions, MolReasoner enhances the accuracy and semantic consistency of the generated molecular descriptions through the introduction of reasoning-enhanced mechanisms, overcoming the limitation of traditional methods that rely solely on the final answer. Particularly in the text-based de novo molecule generation task, MolReasoner strengthens the reasoning process, ensuring that the generated molecules not only adhere to chemical logic but also maintain high consistency in structure and semantics, exhibiting greater reliability. Overall, MolReasoner overcomes the limitations of both prompt-based methods and fine-tuning methods without explicit reasoning methods, achieving stronger generation performance and higher chemical plausibility in both captioning and generation tasks.

| Method | Size | BLEU-2↑ | BLEU-4↑ | METEOR↑ | ROUGE-1↑ | ROUGE-2↑ | ROUGE-L↑ |
|---|---|---|---|---|---|---|---|
| *Closed-Source Model* | | | | | | | |
| GPT-4o | - | 0.1194 | 0.0433 | 0.1651 | 0.2315 | 0.0738 | 0.1792 |
| *Ours* | | | | | | | |
| Warm-up | 7B | 0.3822 | 0.2683 | 0.4185 | 0.4970 | 0.3100 | 0.4304 |
| + FAR | 7B | 0.3853 | 0.2715 | 0.4206 | 0.4999 | 0.3101 | 0.4317 |
| + BLEU2 | 7B | **0.4388** | 0.3190 | 0.4731 | 0.5484 | 0.3598 | 0.4762 |
| + BLEU4 | 7B | 0.4377 | 0.3207 | 0.4751 | 0.5503 | 0.3644 | 0.4799 |
| + METEOR | 7B | 0.4379 | 0.3187 | 0.4752 | 0.5493 | 0.3607 | 0.4756 |
| + ROUGE1 | 7B | 0.4380 | 0.3205 | 0.4752 | 0.5522 | 0.3634 | 0.4786 |
| + ROUGE2 | 7B | 0.4369 | 0.3211 | 0.4711 | **0.5530** | 0.3652 | 0.4805 |
| Zero-RL + $R_{\text{language}}$ | 7B | 0.1339 | 0.0582 | 0.2385 | 0.2782 | 0.1028 | 0.2076 |
| MolReasoner + $R_{\text{language}}$ | 7B | 0.4383 | **0.3220** | **0.4754** | **0.5530** | **0.3662** | **0.4821** |

Table 3: Ablation study of different reward functions and the effect of the warm-up stage for Molecule Captioning. "Warm-up" denotes the base model without reinforcement learning. "FAR" is Format Accuracy Reward; "BLEU2/4," "METEOR," and "ROUGE" are their respective metric rewards. "Zero-RL + $R_{\text{language}}$" trains without warm-up, while "MolReasoner + $R_{\text{language}}$" is the final model. MolReasoner achieves the best performance with $R_{\text{language}}$, highlighting its superiority in captioning.

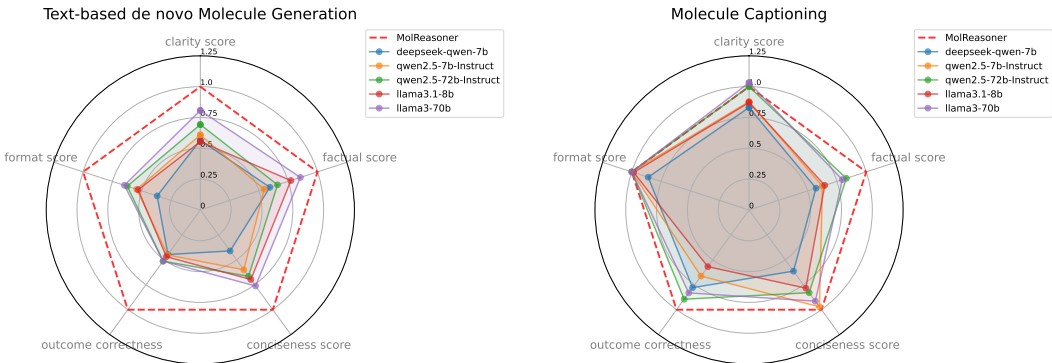

Figure 3: Performance of all models across five key evaluation metrics in the two tasks: Text-based de novo Molecule Generation and Molecule Captioning. To provide a more intuitive comparison, all scores are normalized by dividing them by the scores of MolReasoner.

## 3.4 Multidimensional Evaluation and Qualitative Analysis

To obtain a more comprehensive evaluation of the whole model responses, we developed a multidimensional scoring methodology that assesses both the reasoning process and the final output. This framework extends beyond standard task-specific metrics to include five key dimensions: **Clarity of Logic**, **Factual Correctness**, **Conciseness**, **Format correctness** and **Outcome Correctness**. To implement this framework, we engaged domain experts to manually score a subset of responses, establishing a gold-standard dataset. This expert-annotated data served as few-shot examples in a detailed prompt for the powerful large language model, GPT-5. Through this process, we configured GPT-5 to act as a reliable and scalable automated rater, aligning its scoring with the nuanced judgment of our human experts.

The performance of each model is visualized using a radar map Figure 3, providing an intuitive comparison across five metrics and revealing nuanced performance profiles. This map shows that MolReasoner's superiority lies not just in the accuracy of its final answers, but in its reasoning process. This process is highly explainable and readable, and it demonstrates more precise chemical information recognition than baselines.

## 3.5 Ablation Study

### 3.5.1 Reward Ablation for Molecule Captioning

We systematically evaluate the impact of introducing different stages and reward functions as shown in Table 3. First, introducing a knowledge-guided CoT warm-up already brings substantial improvements, confirming that structured reasoning enhances molecular understanding. Adding Format Accuracy Reward (FAR) further stabilizes and refines outputs. Incorporating BLEU and ROUGE

| Method | BLEU↑ | Exact↑ | Levenshtein↓ | RDK FTS↑ | MACCS FTS↑ | MORGAN FTS↑ | Frag-J↑ | Frag-R↑ | FG-Match↑ | VALIDITY↑ |
|---|---|---|---|---|---|---|---|---|---|---|
| | | | | | *Closed-Source Model* | | | | | |
| GPT-4o | 0.1949 | 0.0045 | 49.3545 | 0.0926 | 0.2066 | 0.0836 | 0.1296 | 0.1777 | 0.3753 | 0.2916 |
| | | | | | *Ours* | | | | | |
| Warm-up | 0.7466 | 0.0652 | 27.2543 | 0.4179 | 0.6601 | 0.3502 | 0.4830 | 0.6047 | 0.5130 | 0.9263 |
| + FAR | 0.7599 | 0.0740 | 27.1335 | 0.4225 | 0.6612 | 0.3516 | 0.4903 | 0.6097 | 0.5174 | 0.9670 |
| + $FP_{sim}$ | 0.7637 | 0.0740 | 27.1249 | 0.4312 | 0.6718 | 0.3566 | 0.5018 | 0.6166 | 0.5181 | 0.9671 |
| + $FRAG_{sim}$ | 0.7637 | 0.0742 | 27.1155 | 0.4294 | 0.6647 | 0.3588 | 0.5151 | 0.6344 | 0.5330 | 0.9674 |
| + $FG_{match}$ | 0.7684 | 0.0746 | 26.9617 | 0.4347 | 0.6754 | 0.3603 | 0.5168 | 0.6375 | 0.5387 | 0.9633 |
| Zero-RL + $R_{structural}$ | 0.1563 | 0.0039 | 34.9341 | 0.1121 | 0.3407 | 0.0990 | 0.2017 | 0.2875 | 0.3167 | 0.7223 |
| MolReasoner + $R_{structural}$ | **0.7841** | **0.0758** | **26.9255** | **0.4373** | **0.6759** | **0.3627** | **0.5213** | **0.6414** | **0.5390** | **0.9679** |

Table 4: Ablation study on different reward functions and the effect of warm-up stages for Text-based de novo Molecule Generation. "FAR" stands for Format Accuracy Reward, "$FP_{sim}$" refers to the fingerprint similarity combining RDK, MACCS, and MORGAN, "$FRAG_{sim}$" refers to the fragment similarity score combining Frag-J and Frag-R, "$FG_{match}$" refers to the functional group matching score, and "Zero-RL" indicates the model trained without warm-up.

rewards then drives consistent gains across language metrics. Notably, even without warm-up, Zero-RL with our language-similarity reward surpasses GPT-4o, underscoring the effectiveness of the reward itself. Finally, MolReasoner—combining warm-up, format accuracy, and multi-level rewards—achieves the best overall performance, validating the synergistic effect of our design.

### 3.5.2 Reward Ablation on Text-based de novo Molecule Generation

In the ablation study for the text-based de novo molecule generation task, we evaluated the impact of introducing different stages and reward functions, comparing the results with baseline models such as GPT-4o. The results in Table 4 show that introducing a knowledge-guided CoT warm-up significantly improves generation quality and reasoning capability. Adding the Format Accuracy Reward (FAR) further refines outputs, while incorporating chemistry-specific rewards such as fingerprint, fragment, and functional group similarity substantially enhances structural fidelity and semantic consistency beyond what format rewards alone can achieve. Even without warm-up, Zero-RL with structural rewards already surpasses GPT-4o in validity. With warm-up, format, and structural rewards combined, MolReasoner outperforms all baselines, yielding more accurate, valid, and chemically meaningful molecule designs.

## 4 Limitations

Despite the strong empirical gains of MolReasoner, several limitations remain. First, our reliance on synthetic CoT rationales generated by GPT-4o may introduce biases and errors, and we lack a calibrated confidence estimate for its reasoning chains. Second, the reward functions, while effective at improving structural similarity and validity, do not account for properties such as synthetic accessibility or 3D conformational feasibility; more comprehensive and experimentally grounded evaluations are needed. Finally, the two-stage fine-tuning and on-policy reinforcement learning pipeline incurs significant computational cost, which may limit scalability to larger models and molecule libraries; optimizing for efficiency will be a key direction for our future work.

## 5 Conclusion

In this paper, we introduce MolReasoner, a two-stage training framework designed to advance large language models (LLMs) beyond memorized knowledge and toward effective reasoning in molecular tasks. Experimental results demonstrate that the integration of supervised fine-tuning (Mol-SFT) with reinforcement learning (Mol-RL) overcomes the limitations of rote memorization, significantly improving accuracy, interpretability, and structural understanding. The ability to translate between molecular structures and human language reveals vast potential to transform fields such as drug discovery, chemical synthesis, and materials science, paving the way for a future where the interface between LLMs and reasoning drives scientific innovation.

# 6 ETHICS STATEMENT

In developing MolReasoner, we prioritized ethical considerations to ensure the responsible use of our models and methodologies. First, this research does not involve human subjects, and all datasets used (e.g., ChEBI-20) are publicly available and copyright-compliant. We applied strict data filtering to guarantee chemical validity and minimize the risk of introducing biased or misleading molecule–text pairs. Nevertheless, we acknowledge that biases inherent in benchmark datasets (e.g., underrepresentation of certain molecular families) may propagate into the model's outputs. We adhere to all relevant legal and ethical research guidelines, including respecting open-source licenses during dataset construction and providing comprehensive model documentation. Our work is conducted with a strong commitment to research integrity, ensuring that our contributions remain beneficial to the scientific community and the AI for Science domain while addressing the ethical responsibilities associated with molecular AI technologies.

# 7 REPRODUCIBILITY STATEMENT

We have made extensive efforts to ensure the reproducibility of our work. The proposed MolReasoner training pipeline is described in detail in the main text and appendix, including the Mol-SFT and Mol-RL stages. Details of dataset construction, reward design, and evaluation metrics are provided in the main text and appendix. We provide the code in the supplement material. Through these efforts, we aim to enable future research to reliably reproduce, validate, and extend our findings.

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

## A  THE USE OF LARGE LANGUAGE MODELS

The authors declare that the human authors are the sole contributors to this work. This paper was written and edited exclusively by the authors. Large Language Models (LLMs) were used solely as a general-purpose tool to aid in the writing and editing process. Specifically, an LLM was utilized for:

- **Text Polishing**: Improving the grammar, syntax, and flow of certain paragraphs to enhance overall readability.

- **Data Augmentation and Model Evaluation**: Assisting in generating synthetic data for supervised fine-tuning and served as a tool for evaluating model responses. Detailed prompts and methods for this data synthesis are provided in the main paper and in the 'Prompts' section of the appendix. All data generated or evaluated with the assistance of the LLM were thoroughly verified by the authors.

The authors have reviewed, edited, and verified all content generated or augmented by the LLM and take full responsibility for the entire contents of the paper. The use of LLMs does not constitute authorship.

## B  RELATED WORK

In this section, we provide a review of literature related to molecular language models and large reasoning models.

### B.1  MOLECULAR LANGUAGE MODELS

Early approaches to molecular understanding represent molecules as 1D sequences. KV-PLM Zeng et al. (2022) leverages SMILES Weininger (1988) to represent molecules and employs a masked-language-modeling objective for pretraining on biomedical texts. MolT5 Edwards et al. (2022), based on T5 Raffel et al. (2020), is specifically designed for molecular translation tasks. More recently, LlaSMol Yu et al. (2024) fine-tunes a suite of open-source LLMs on self-curated molecular instruction datasets. Mol-Instructions Fang et al. (2024) adopts the SELFIES Krenn et al. (2022) molecular descriptor and introduces a dedicated instruction dataset for biomolecular research. The introduction of molecular graph encoders has led to the development of multimodal molecular language models. MoMu Su et al. (2022) and MoleculeSTM Liu et al. (2022) employ cross-modal contrastive learning to bridge the representation spaces of molecular graphs and text. MolCA Liu et al. (2023) combines SMILES with 2D molecular representations for molecule-to-text generation. More recent work, such as 3D-MoLM Li et al. (2024) and BioT5+ Pei et al. (2024), incorporates 3D molecular structures to enhance LLMs' ability to model molecular understanding. Despite these advancements, these models remain limited in their reasoning capabilities due to the absence of Chain-of-Thought (CoT) Wei et al. (2022) fine-tuning. This gap restricts their performance in tasks requiring complex molecular understanding and reasoning, thereby limiting their practical utility in more demanding biomedical applications.

### B.2  LARGE REASONING MODELS

Recent advancements have led to the emergence of Large Reasoning Models (LRMs) Jaech et al. (2024); Team et al. (2023); Liu et al. (2024a), which extend the capabilities of traditional LLMs by enabling deliberative, multi-step reasoning. These models distinguish themselves through the explicit representation of reasoning processes, which is crucial for tackling complex tasks requiring structured problem-solving. The development of LRMs is closely tied to policy optimization techniques for model alignment, with the canonical approach being Reinforcement Learning from Human Feedback (RLHF) Ouyang et al. (2022) with Proximal Policy Optimization (PPO) Schulman et al. (2017). However, the computational demands and complexity of this approach, which involves managing multiple models (*i.e.,* policy, reference, reward, and critic), have driven the exploration of more efficient alternatives. One such alternative is Group Relative Policy Optimization (GRPO), introduced in the training of DeepSeekMath Shao et al. (2024). GRPO is a variant of PPO

that eliminates the need for a separate critic network, thereby reducing both memory and computational overhead. The efficiency and versatility of GRPO have enabled its application in a variety of high-stakes domains beyond its initial focus on mathematics, including puzzles Xie et al. (2025), medicine Lai et al. (2025), and finance Liu et al. (2025). Despite the progress in LRM development, their application to molecular tasks remains relatively underexplored, presenting a promising avenue for this paper.

## C MOLECULE-TEXT TRANSLATION

Advancing the use of LLMs in molecular science requires moving beyond structural representations toward explicit structure-level reasoning grounded in natural language. Such reasoning abilities are essential for interpreting molecular semantics, inferring molecular functions, and generating chemically plausible structures from textual inputs. To systematically evaluate these capabilities, researchers Edwards et al. (2022) introduced the *molecule-text translation* task, designed to assess a model's proficiency in aligning and reasoning between molecular representations and natural language descriptions. This task comprises two complementary sub-tasks:

- **Molecule Captioning**: Given a molecular descriptor (*e.g.,* SMILES Weininger (1988), SELFIES Krenn et al. (2022), or IUPAC name Favre & Powell (2013)), the objective is to generate a coherent natural-language description capturing structural characteristics, functional roles, and potential applications of the molecule. This evaluates the model's ability to interpret molecular structures and abstract their semantics into text.

- **Text-based de novo Molecule Generation**: Conversely, this task requires the model to generate valid molecular descriptors from natural language descriptions, testing its capacity to map textual semantics to relevant chemical motifs, and produce syntactically and chemically valid molecular structures.

Previous studies Taylor et al. (2022); Edwards et al. (2022) uses SMILES for molecule representation, its grammar and token order sensitivity often result in invalid or chemically implausible outputs. primarily employed SMILES for molecular representation. However, the grammar and token-order sensitivity inherent in SMILES frequently lead to invalid or chemically implausible outputs. To overcome this limitation, we adopt SELFIES representations, following the Mol-Instructions framework Fang et al. (2024). SELFIES ensures chemical validity by construction, eliminating common structural errors such as mismatched parentheses, invalid atomic symbols, and illogical branching patterns.

### C.1 GROUP RELATIVE POLICY OPTIMIZATION

The core innovation of Group Relative Policy Optimization (GRPO) Shao et al. (2024) lies in its group-based redefinition of the advantage function. In contrast to PPO, GRPO removes the value function and estimates advantages relative to a sampled response group. Specifically, given a question-answer pair $(q, a)$, the old policy $\pi_{\theta_{\text{old}}}$ samples a group of $G$ responses $\{o_i\}_{i=1}^{G}$. The advantage for the $i$-th response is computed by normalizing the corresponding group-level rewards $\{R_i\}_{i=1}^{G}$:

$$\hat{A}_{i,t} = \frac{r_i - \text{mean}(\{R_i\}_{i=1}^{G})}{\text{std}(\{R_i\}_{i=1}^{G})}. \tag{8}$$

Additionally, GRPO employs a clipped objective combined with a KL-divergence penalty term, defined as:

$$\mathcal{J}_{\text{GRPO}}(\theta) = \mathbb{E}_{(q,a)\sim\mathcal{D},\{o_i\}_{i=1}^{G}\sim\pi_{\theta_{\text{old}}}(\cdot|q)} \left[ \frac{1}{G}\sum_{i=1}^{G}\frac{1}{|o_i|}\sum_{t=1}^{|o_i|} \left( \right. \right.$$

$$\min\left(r_{i,t}(\theta)\hat{A}_{i,t}, \text{ clip}\left(r_{i,t}(\theta), 1-\varepsilon, 1+\varepsilon\right)\hat{A}_{i,t}\right) \tag{9}$$

$$\left. \left. - \beta D_{\text{KL}}(\pi_\theta \parallel \pi_{\text{ref}}) \right) \right]$$

where the importance ratio $r_{i,t}(\theta)$ is given by:

$$r_{i,t}(\theta) = \frac{\pi_\theta(o_{i,t} \mid q, o_{i,<t})}{\pi_{\theta_{\text{old}}}(o_{i,t} \mid q, o_{i,<t})}. \tag{10}$$

## D   TRAINING SETUP.

We use Qwen2.5-7B-Instruct as the base model. During the warm-up stage, we fine-tune the model separately on CoT data for both tasks. The learning rate is set to $1 \times 10^{-5}$, with a total of 4 epochs, and a cosine learning rate scheduler is applied to stabilize training. Following warm-up, we perform GRPO reinforcement learning. In this stage, the batch size is 256, the learning rate is $1 \times 10^{-7}$, and each sample generates 8 candidate outputs (rollouts). A temperature of 0.7 is applied to encourage output diversity. We train the models for 15 epochs to fully optimize generation performance across both tasks. All experiments are run on 8 Tesla-A100-80G GPUs. For molecule captioning and text-based de novo molecule generation, Mol-SFT requires approximately 1 GPU hour. Mol-RL requires approximately 1200 GPU hours.

## E   PROMPTS

In this section, we present the input prompts used for knowledge-guided chain-of-thought data construction with GPT-4O, and also the prompts used to evaluate the quality of the model responses. These prompts cover both the tex-based de novo molecule generation and the molecule captioning tasks.

---

**Example 1: Molecule captioning task**

You are a professional chemist. Given a molecule's **SELFIES** and **structural features**, and identified **functional fragments**, your task is to generate a natural, concise, and chemically accurate description of the molecule. Do not reproduce or reference the original SELFIES string. Instead, decode it internally and abstractly summarize the key structural features it represents.
**SELFIES**:
`[O][=C][Branch1][C][O-1][C][C][S][S][C][Ring1][Branch1]`
**Structural Info**:
1. The molecule has 1 ring(s), including 0 aromatic ring(s).
2. The molecular weight is approximately 149.22 g/mol.
**Fragments**:
`<|O=C[O-]|><|SS|><|C030|><|C020|><|C020|>`
Please provide a step-by-step analysis that explains how you would interpret this molecular structure and summarize it into a concise and chemically accurate natural language description. Let's think step by step and return the final answer in <answer> The molecule is a monocarboxylic acid anion and a member of dithiolanes. It is a conjugate base of an asparagusic acid. It derives from a hydride of a 1,2-dithiolane. </answer> tags.

---

**Example 2: Text-based molecule generation task**

You are a professional biochemist designing molecular structures. Given the **molecular description**, basic **structural information**, and identified **fragments**. Your goal is not to re-predict the molecule's SELFIES, but to generate a logical, chemically sound reasoning chain that explains how one could deduce or construct this structure based on the given information.
**Description**:
1. The molecule is an aldehyde that is thiphene substituted by a formyl group at position.
2. It has a role as a metabolite. It is a member of thiophenes and an aldehyde
**Structural Info**:
1. The molecule has 1 ring(s), including 1 aromatic ring(s).
2. The molecular weight is approximately 112.15 g/mol.
**Fragments**:
`<|C=O|><|c1ccsc1|>`

---

Please provide a step-by-step molecular reasoning chain that explains how you would reconstruct or deduce the molecular structure. Let's think step by step and return the final answer in `<answer> [O][=C][C][=C][C][=C][S][Ring1][Branch1] </answer>` tags.

### Example 3: Example for quality evaluation of the model responses

You are a professional chemist and an expert evaluator. Your task is to assess and score a molecular description (Chain of Thought, or CoT) generated by an AI model, based on specific criteria. Your evaluation must be strict, objective, and consistent with the provided examples. Please score the CoT on a scale of 1-5 across three key dimensions:

#### 1. CLARITY OF LOGIC & INSIGHTFULNESS

This criterion evaluates whether the CoT's reasoning process is coherent and if it provides valuable chemical insights.

- **5 (Outstanding)**: The reasoning is exceptionally clear, with a flawless logical progression. The CoT provides profound chemical insights beyond simple facts, demonstrating expert-level thought.
- **4 (Excellent)**: The logical flow is very clear and the analysis is systematic. It provides accurate insights but may lack particularly novel or deep observations.
- **3 (Acceptable)**: The reasoning is mostly clear, but there may be minor logical jumps. The insights provided are correct but basic.
- **2 (Lacking)**: The reasoning is disorganized or difficult to follow. The connection between analysis and conclusions is weak.
- **1 (Poor)**: There is no recognizable logic or insight. The CoT is a disorganized list of facts with no meaningful analysis.

#### 2. FACTUAL CORRECTNESS

This criterion evaluates the accuracy of all factual statements within the CoT.

- **5 (Completely Accurate)**: All chemical statements, nomenclature, and factual details are entirely correct with no inaccuracies.
- **4 (Minor Errors)**: Contains one or two subtle, inconsequential errors that do not affect the overall conclusion.
- **3 (Partially Accurate)**: Contains a few identifiable factual errors or inaccuracies that do not fundamentally break the reasoning.
- **2 (Multiple Errors)**: Contains several clear and misleading factual errors.
- **1 (Severely Flawed)**: Riddled with serious factual errors, making the entire analysis untrustworthy.

#### 3. REDUNDANCY & CONCISENESS

This criterion measures the efficiency of the CoT. A high-quality CoT should contain only necessary steps, avoiding repetition.

- **5 (Extremely Concise)**: Every step in the CoT is essential. There are no redundant sentences or repeated analyses; the text is efficient and to the point.
- **4 (Concise)**: Most steps are necessary. There may be one or two sentences that could be trimmed, but the overall text is not redundant.
- **3 (Acceptable)**: Contains some redundant information that could be merged or removed, but the overall structure remains clear.
- **2 (Verbose)**: Contains repetitive analysis or unnecessary information that makes the text feel bloated.

- **1 (Extremely Verbose)**: The text is filled with a large amount of repetitive or irrelevant content, making it difficult to read.

---

**Important Note: Adjusting for Scoring Bias**
Please be aware that human experts tend to be more lenient and generous in their scoring than you. When a CoT is of high quality, experts often assign scores of **4 or 5** even if there are minor imperfections. In contrast, your current scoring may be too conservative.
When performing your evaluation, please adjust your internal scoring scale to align with this more generous, expert-like style. For high-quality CoTs, **do not hesitate to assign 4s and 5s**.

---

Your final evaluation must be returned as a **JSON object**, and it must contain only this JSON. The JSON must include the following keys:

- `clarity_score`: The score for Clarity of Logic & Insightfulness (1-5)
- `factual_score`: The score for Factual Correctness (1-5)
- `conciseness_score`: The score for Redundancy & Conciseness (1-5)

### FEW-SHOT EXAMPLES

**Follow these examples strictly, mimicking the format and scoring style:**
**Example 1:**
...
**Human expert's score:** { "clarity score": 5, "factual score": 4, "conciseness score": 4, }
**Example 2:**
...
**Human expert's score:** { "clarity score": 3, "factual score": 5, "conciseness score": 4, }
**Example 3:**
...
**Human expert's score:** { "clarity score": 4, "factual score": 3, "conciseness score": 5, }
Please rate the following model response:

## F  SUCCESS AND FAILURE CASES

In this section, we present representative success and failure cases for both molecule generation and molecule captioning tasks, with some basic analysis of their respective error modes. Table 5 and 6 show the model responses for molecule generation, while Table 7 and 8 show those for molecule captioning.

Table 5: **Case I used to illustrate the model responses of the molecule generation task.** The top block shows the prompt used to test the models, and the following blocks show the response of prompt-based method(Qwen2.5-72B-Instruct), the model after fine-tuning without explicit reasoning and our method.

---

**Prompt**

You are a professional biochemist designing molecular structures. Please generate the molecular structure (SELFIES) based on the following description. Please **think step by step** and return the final answer in <answer> ... </answer> tags.

The molecule is a phosphatidylcholine 32:0 in which the 1- and 2-acyl groups are specified as hexadecanoyl (palmitoyl). A synthetic phospholipid used in liposomes and lipid bilayers to study biological membranes. It is also a major constituent of pulmonary surfactants. It has a role as a surfactant and a mouse metabolite. It is a phosphatidylcholine 32:0 and a 1-acyl-2-hexadecanoyl-sn-glycero-3-phosphocholine. It derives from a hexadecanoic acid.

**Ground truth**: [C][C][C][C][C][C][C][C][C][C][C][C][C][C][C][C][=Branch1][C][=O][O][C][C@H1][Branch2][Ring1][Branch1][C][O][P][=Branch1][C][=O][Branch1][C][O-1][O][C][C][N+1][Branch1][C][C][Branch1][C][C][C][O][C][=Branch1][C][=O][C][C][C][C][C][C][C][C][C][C][C][C][C][C]

---

**Prompt-based method: Qwen2.5-72B-Instruct**

To generate the SELFIES (SELF-referencING mOLECULAR stringS) for the phosphatidylcholine 32:0 with 1- and 2-acyl groups specified as hexadecanoyl (palmitoyl), we need to break down the structure step by step.

Step-by-Step Breakdown

1. Hexadecanoyl (Palmitoyl) Chain: Hexadecanoyl is a 16-carbon saturated fatty acid. SELFIES for hexadecanoyl: [C][C][C][C][C][C][C][C][C][C][C][C][C][C][C][C][O][=O]

2. Glycerol Backbone: Glycerol is a three-carbon alcohol. SELFIES for glycerol: [C][O][C][O][C][O]

3. Phosphate Group: The phosphate group is attached to the third carbon of glycerol. SELFIES for phosphate: [P][O][O][O][O]

4. Choline Head Group: Choline is a quaternary ammonium compound. SELFIES for choline: [N][C][C][C][O][C][C][C][C][C][C][C]...(approx. 100 [C])[C][C]

**(Hallucinated structure. Repetitive fragments. Format not followed.)**

---

**Model after fine-tuning Without Explicit Reasoning**

[C][C][C][C][C][C][C][C][C][C][C][C][C][C][C][C][=Branch1][C][=O][O][C][C@H1][Branch2][Ring1][Branch1][C][O][P][=Branch1][C][=O][Branch1][C][O][O][C][C][N+1][Branch1][C][C][Branch1][C][C][C][O][C][=Branch1][C]

**(Valid but incorrect molecule. No reasoning. Format not followed.)**

---

**Our method**

<think>

To deduce the molecular structure of the phosphatidylcholine 32:0 with specified acyl groups, we can follow a logical reasoning chain based on the provided description, structural information, and fragments. Here's a step-by-step breakdown:

**1. Identify the Core Structure**: Phosphatidylcholine is a type of ...

**2. Determine the Glycerol Backbone**: The glycerol backbone is a three-carbon molecule with ...

**3. Acyl Groups**: The description specifies that the 1- and 2-acyl groups are hexadecanoyl (palmitoyl). Hexadecanoyl refers to a fatty acid with 16 carbon atoms (C16). Therefore, ...

**4. Phosphocholine Head Group**: The third carbon of the glycerol is attached to a phosphocholine group. This group ...

... (omitted structural construction and molecular weight calculation steps)

**8. Use the Provided Fragments**: The fragments given in the description can be used to construct the molecular structure. For example, the fragment ...

... (omitted rest of structural deducing reasoning, less central to SELFIES construction)

The final answer, representing the constructed molecular structure, is:

<answer>[C][C][C][C][C][C][C][C][C][C][C][C][C][C][C][C][=Branch1][C][=O][O][C][C@H1][Branch2][Ring1][Branch1][C][O][P][=Branch1][C][=O][Branch1][C][O-1][O][C][C][N+1][Branch1][C][C][Branch1][C][C][C][O][C][=Branch1][C][=O][C][C][C][C][C][C][C][C][C][C][C][C][C]</answer>

**(Clear reasoning. Valid and correct molecule output. Desired format.)**

---

972
973
974
975
976
977
978
979
980
981
982
983
984
985
986
987
988
989
990
991
992
993
994
995
996
997
998
999
1000
1001
1002
1003
1004
1005
1006
1007
1008
1009
1010
1011
1012
1013
1014
1015
1016
1017
1018
1019
1020
1021
1022
1023
1024
1025

Table 6: **Case II used to illustrate the model responses of the molecule generation task.** The top block shows the prompt used to test the models, and the following blocks show the response of prompt-based method(DeepSeek-R1-Distill-Qwen-7B), the model after fine-tuning without explicit reasoning and our method.

**Prompt**

You are a professional biochemist designing molecular structures. Please generate the molecular structure (SELFIES) based on the following description. Please **think step by step** and return the final answer in <answer> ... </answer> tags.

The molecule is a 1-O-acyl-sn-glycero-3-phosphocholine in which the acyl group is specified as caproyl (hexanoyl). It is a 1-O-acyl-sn-glycero-3-phosphocholine, a hexanoate ester and a lysophosphatidylcholine 6:0.

**Ground truth**: [C][C][C][C][C][C][=Branch1][C][=O][O][C][C@@H1][Branch1][C][O][C][O][P][=Branch1][C][=O][Branch1][C][O-1][O][C][C][N+1][Branch1][C][C][Branch1][C][C][C]

---

**Prompt-based method: DeepSeek-R1-Distill-Qwen-7B**

<think>

Okay, so I need to figure out how to construct the SELFIES notation for this molecule. Let me start by understanding the structure. The molecule is ... But in SELFIES, it's represented as [C][C@H1][C@@H1][Branch2]. Hmm, maybe I should break it down....(another round of reasoning)The hexanoyl would be a chain of six carbons with an ester group, so maybe [C][C][C](O)=O connected to the hexanoyl.

Wait, perhaps it's better to structure it step by step. The choline part is ...(reasoning to construct the SELFIES)I think the correct SELFIES would be: [C][C@H1][C@@H1][Branch2]O[C][C][C](O)=O[C][C][C]. Wait, that might not be accurate. Let me make sure.Alternatively, maybe it's [C][C@H1][C@@H1][Branch2]O[C][C][C](O)=O[C][C][C]. No, that doesn't seem right because ... Putting it all together, the SELFIES should be: [C][C@H1][C@@H1][Branch2]O[C][C][C](O)=O[C][C][C]. Wait, no, the hexanoyl is ... Hmm, I'm getting confused. Let me try to write it step by step.1. Choline: [C][C@H1][C@@H1][Branch2]... So, perhaps: [C][C@H1][C@@H1][Branch2]O[C][C][C](O)=O[C][C][C][C][C][C]

Wait, that still doesn't look right. Maybe I should ... Putting it all together, the SELFIES would be:[C][C@H1][C@@H1][Branch2]O[C][C][C](O)=O[C][C][C][C][C][C]

Wait, no, the hexanoyl should be... The glycerol's C2 has the hexanoyl: [C][C][C](O)=O.

Wait, perhaps SELFIES is:[C][C@H1][C@@H1][Branch2]O[C][C][C](O)=O[C][C][C][C][C][C][C].

But I'm not confident. Maybe I should represent the glycerol as ... Putting it all together, the SELFIES would be:[C][C@H1][C@@H1][Branch2]O[C][C][C](O)=O[C][C][C][C][C][C][C]. Wait, that seems too long.

**(Frequent self-doubt. Format not followed.)**

---

**Model after fine-tuning Without Explicit Reasoning**

[C][C][C][C][C][C][=Branch1][C][=O][O][C][C@H1][Branch2][Ring1][Branch1][C][O][P][=Branch1][C][=O][Branch1][C][O][O][C][C][N+1][Branch1][C][C][Branch1][C][C][C][O]

**(Lack of Chain-of-Thought. Format not followed. Generally accurate yet deviates subtly from the ground truth.)**

---

**Our method**

<think>

To deduce the molecular structure of 1-O-acyl-sn-glycero-3-phosphocholine with a caproyl (hexanoyl) acyl group, we can follow a logical reasoning chain based on the provided description, structural information, and fragments. Here's a step-by-step breakdown:

1. **Identify the Core Structure**: The molecule is a type of phospholipid, specifically ...

2. **Acyl Group**: The acyl group is specified as caproyl, which is a straight-chain fatty acid with six carbon atoms (hexanoate). This means the acyl group is -C(=O)CH2CH2CH2CH2CH3.

3. **Glycerol Backbone**: The glycerol backbone consists of three carbon atoms, where ...

4. **Phosphocholine Group**: The phosphocholine group consists of a phosphorus atom bonded to ...

5. **Construct the Molecular Structure**: Start with the glycerol backbone: C1 is attached to ...

6. **Verify the Molecular Weight**: The molecular weight of the constructed molecule should be ...

7. **Check the Chemical Connectivity**: Ensure that all atoms are connected correctly and ...

8. **Finalize the Structure**: Once the structure is constructed and verified, it can be represented in a standardized format, such as SELFIES or a similar notation.

The final answer, representing the constructed molecular structure, is:

<answer>[C][C][C][C][C][C][=Branch1][C][=O][O][C][C@@H1][Branch1][C][O][C][O][P][=Branch1][C][=O][Branch1][C][O-1][O][C][C][N+1][Branch1][C][C][Branch1][C][C][C] </answer>

**(Clear and transparent reasoning. Valid and correct molecule output. Desired format.)**

Table 7: **Case I used to illustrate the model responses of the molecule captioning task.** The top block shows the prompt used to test the models, and the following blocks show the response of prompt-based method(GPT-4O), the model after fine-tuning without explicit reasoning and our method.

---

**Prompt**

You are a professional chemist. Your task is to generate a natural, concise, and chemically accurate description of a given molecule. Please provide a **step-by-step** analysis explaining how you interpret the molecular structure, identify key features and functional groups, and summarize it into a clear and informative description. Think step by step, and your final answer **must** be returned in the format: <answer> ... </answer>. For example:

<answer>The molecule is an epoxy(hydroxy)icosatrienoate that is the conjugate base of 11-hydroxy-(14R,15S)-epoxy-(5Z,8Z,12E)-icosatrienoic acid, obtained by deprotonation of the carboxy group; major species at pH 7.3. It is a conjugate base of an 11-hydroxy-(14R,15S)-epoxy-(5Z,8Z,12E)-icosatrienoic acid.</answer>

[C][S][C][C][C@H1][Branch1][C][NH3+1][C][=Branch1][C][=O][N][C@@H1][Branch1]
[C][C][C][=Branch1][C][=O][O-1]

Label: The molecule is a peptide zwitterion obtained by transfer of a proton from the carboxy to the amino terminus of Met-Ala. It is a tautomer of a Met-Ala.

---

**Prompt-based method: GPT-4O**

To interpret and generate a precise description of the given molecule from its SELFIES notation, follow these analytical steps:

Step 1: Understand SELFIES Notation SELFIES (Self-referencing Embedded Strings) is a high-level representation of molecular structures. Each character or symbol corresponds to ...

Step 2: Parse and Identify Atoms [C]: Represents carbon atoms. [S]: Represents a sulfur atom. [C@H1]: A chiral carbon ...

Step 3: Build the Backbone: By interpreting the sequence, the molecule exhibits a linear carbon chain with branching points, functional groups (amino and carboxylate), and stereochemical centers.

Step 4: Identify Functional Groups and Stereochemistry: An amino group [NH3+1] exists as a protonated NH3+. Two carbonyl groups (=O) are present 2014one possibly as part of an amide and the other as a carboxylate. The sulfur atom [S] suggests a thioether or thioamide moiety.

Step 5: Summarize the Molecule: Integrating stereochemistry, charges, and functional groups results in the clear identification of the compound.

<answer> The molecule is a sulfinyl-dipeptide derivative containing a protonated amino group at one terminus and a negatively charged carboxylate group at the other. It includes a sulfur atom within the backbone structure, flanked by two stereogenic centers marked with opposing stereochemistry (R and S configurations). The carbon skeleton features both an amide linkage and carboxylate functionality, consistent with its zwitterionic form at physiological pH (7.3).</answer>

**(Inaccurate SELFIES parsing. Overgeneralized functional group assignments. Unsupported structural assumptions.)**

---

**Model after fine-tuning Without Explicit Reasoning**

The molecule is zwitterionic form of N(6),N(6),N(6)-trimethyl-L-lysine arising from transfer of a proton from the carboxy to the amino group; major species at pH 7.3 It is a tautomer of a N(6),N(6),N(6)-trimethyl-L-lysine.

**(Lack of Chain-of-Thought. Format not followed.)**

---

**Our method**

<think>

To interpret the provided molecular structure based on the SELFIES, structural information, and functional fragments, we can follow these steps:

**1. Identify the Core Structure**: The molecule appears to be a peptide, specifically a dipeptide, consisting of ...

**2. Analyze the Functional Groups**: The key functional groups include: An amino group (NH3+), which is ...

**3. Examine the Amino Acid Residues**: The structure includes L-methionine and L-alanine: L-methionine contributes ...

**4. Consider the Zwitterionic Nature**: At a pH of 7.3, the amino group is likely ...

**5. Summarize the Findings:** Based on the analysis, we can conclude that the molecule is a ...

Now, we can formulate a concise and chemically accurate description of the molecule:

<answer>The molecule is a peptide zwitterion obtained by transfer of a proton from the carboxy to the amino group of Met-Ala; major species at pH 7.3. It is a tautomer of a Met-Ala. </answer>

**(Clear reasoning. Valid and correct caption output. Desired format.)**

Table 8: **Case II used to illustrate the model responses of the molecule captioning task.** The top block shows the prompt used to test the models, and the following blocks show the response of prompt-based method(Qwen2.5-72B-Instruct), the model after fine-tuning without explicit reasoning and our method.

---

**Prompt**

You are a professional chemist. Your task is to generate a natural, concise, and chemically accurate description of a given molecule. Please provide a step-by-step analysis explaining how you interpret the molecular structure, identify key features and functional groups, and summarize it into a clear and informative description. Think step by step, and your final answer must be returned in the format: <answer> ... </answer>. For example:

<answer>The molecule is an epoxy(hydroxy)icosatrienoate that is the conjugate base of 11-hydroxy-(14R,15S)-epoxy-(5Z,8Z,12E)-icosatrienoic acid, obtained by deprotonation of the carboxy group; major species at pH 7.3. It is a conjugate base of an 11-hydroxy-(14R,15S)-epoxy-(5Z,8Z,12E)-icosatrienoic acid.</answer>

[C][S][C][C][C@H1][Branch1][S][N][C][=Branch1][C][=O][C@@H1][Branch1][C][N][C][C][Branch1][C][C][C][C][=Branch1][C][=O][O]

**Label**: The molecule is a dipeptide formed from L-leucine and L-methionine residues. It has a role as a metabolite. It derives from a L-leucine and a L-methionine.

---

**Prompt-based method: Qwen2.5-72B-Instruct**

**Step-by-Step Analysis**:

Step 1: Identify the Backbone Structure. The given structure is a linear chain of carbon atoms with several branches and functional groups. The main chain can be identified as a series of carbon atoms connected by single and double bonds.

Step 2: Identify Functional Groups. Carbonyl Groups: There are two carbonyl groups (C=O) present in the structure. Amine Groups: There are two amine groups (N) ...

Step 3: Determine Stereochemistry C@H1: Indicates a chiral center with the R configuration...

Step 4: Summarize the Molecule. Combining the information from the steps above, we can describe the molecule as follows:

<answer>The molecule is a branched, chiral compound with a linear carbon backbone. It contains two carbonyl groups, two amine groups, and sulfur atoms. The chiral centers are configured as (R) and (S) at the specified positions. The structure suggests a complex organic molecule with potential biological activity due to its functional groups and stereochemistry.</answer>

**Misidentification of the molecule.Lack of treatment of charge states or protonation. Overly vague final description.**

---

**Model after fine-tuning Without Explicit Reasoning**

The molecule is a L-histidine derivative that is N(alpha)-acetyl-L-histidine substituted by a methyl group at position 5 on the imidazole ring. It has a role as a human urinary metabolite. It is a L-histidine derivative and a N(alpha)-acetyl-L-histidine. It is a conjugate acid of a N(alpha)-acetyl-N(epsilon)-methyl-L-histidine(1-).

**(Lack of Chain-of-Thought. Format not followed. Misidentification of molecule)**

---

**Our method**

<think>

To generate a concise and chemically accurate description of the molecule based on the provided SELFIES, structural features, and functional fragments, we can follow these steps:

1. **Identify the Core Structure**: The molecule appears to be a peptide, specifically a dipeptide, given the presence of amino acid fragments and the presence of an amine (N) and a carboxylic acid (C=O and O).

2. **Analyze the Functional Groups**: The presence of sulfur (S) atoms suggests the inclusion of methionine, an amino acid containing a sulfur atom. The presence of amine (N) and carboxylic acid (C=O and O) groups confirms the peptide nature of the molecule.

3. **Determine the Amino Acid Composition**: The fragments indicate the presence of L-leucine and L-methionine. L-leucine is recognized by the branched carbon structure, while L-methionine is identified by the sulfur atom.

4. **Consider the Stereochemistry**: The notation (C@H1) and (C@@H1) indicates that there are chiral centers in the molecule, which is typical for amino acids.

5. **Summarize the Findings**: Based on the analysis, we can summarize the key features of the molecule, including its composition, structure, and biological relevance.

Now, we can formulate the final answer:

<answer>The molecule is a dipeptide composed of L-leucine and L-methionine joined by a peptide linkage. It has a role as a metabolite. It derives from a L-leucine and a L-methionine. </answer>

**(Clear reasoning. Valid and correct caption output. Desired format.)**

