# OpenReview forum: "MolReasoner: Toward Effective and Interpretable Reasoning for Molecular LLMs"
_ICLR.cc/2026/Conference — ICLR 2026 Conference Withdrawn Submission_

### Official Review · Reviewer_Z9Cn · 2025-10-29

**Soundness:** 2
**Presentation:** 2
**Contribution:** 2
**Rating:** 2
**Confidence:** 4

**Summary:**

This paper proposes MolReasoner, a two-stage framework for molecule-text translation tasks. The approach uses (1) supervised fine-tuning on GPT-4o-distilled Chain-of-Thought trajectories enriched with structural information (Mol-SFT), and (2) reinforcement learning with GRPO using multi-level rewards (Mol-RL). The authors claim to shift LLMs from "memorization" to "reasoning" on molecular tasks, evaluating on ChEBI-20's molecule captioning and text-to-molecule generation tasks. The paper reports metric improvements across both tasks using several different measures.

**Strengths:**

- **Important problem**: Interpretable chemical reasoning is crucial for trustworthy molecular AI applications
- **Well-written**: Clear motivation and comprehensive related work section
- **Reasonable approach**: Two-stage SFT→RL pipeline aligns with recent successful paradigms
- **Domain adaptation**: Incorporating structural features and functional groups into CoT generation is appropriate
- **Expert involvement**: Chemistry experts used in evaluation (Section 3.4), though details are limited
- **Good documentation**: Appendix provides all prompts and case studies, facilitating reproduction

**Weaknesses:**

**Limited novelty**: Training LLMs to reason on chemistry has been shown by Ether0 using the same SFT→RL strategy [6]. This approach (CoT distillation + GRPO) has been applied by DeepSeek and others. The contribution is primarily dataset engineering and reward design, not methodological innovation.

**Flawed motivation (L46-51)**: Claims that general LLMs can't handle molecules lack empirical support and contradict evidence from recent benchmarks (ChemBench [1], ChemEval [2], ChemIQ [3], GPQA chemistry [4], TOMG[5]) showing non-trivial performance. Only anecdotal evidence provided (Figure 1), no systematic evaluation.

**Memorization vs. reasoning poorly operationalized (L52+)**: Knowledge in weights is valid if answers are correct. CoT can be unfaithful/post-hoc rationalization, making explicit reasoning potentially misleading [7, 9]. No evidence that CoT improves generalization vs. answer-only training when controlling for compute.

**GRPO inappropriate for captioning**: GRPO designed for verifiable rewards, but molecule captioning lacks ground truth (multiple valid captions exist). Not justified why GRPO fits this ambiguous setting where only comparison to gold answer may be flawed.

**Misleading naming**: "Mol-SFT" and "Mol-RL" are standard SFT and RL, not novel methods. Suggests algorithmic innovation where contribution is dataset and reward engineering.

**Zero error bars/significance tests**: Table 2 shows baseline exact=0.0737 vs. MolReasoner=0.0758 (2.8% difference) - is this significant? No confidence intervals, multiple runs, or p-values anywhere. Impossible to assess if improvements are real or noise.

**Marginal gains beyond FAR (Tables 3 & 4)**: Format Accuracy Reward provides most improvement. Additional rewards (FP, SELFIES, fragment, FG) add only ~2-5%. Table 4: exact match 0.0740→0.0758 (2.4% gain) - without error bars, likely not significant. The formatting reward alone seems to drive results, suggesting just bad baseline evaluation. Questions whether complex reward design (Equations 2-7) is justified.

**Circular reasoning (L240-257)**: (1) Observe hallucinations, (2) Design metrics to detect them (Frag-J, Frag-R, FG-Match) and rewards to mitigate them, (3) Report improvements on those same metrics. This is circular validation, not rigorous evaluation. Introduction of arbitrary metrics without truly better results.

**Missing critical ablations**: SFT on same data with/without CoT annotations + RL; Evidence that rewards beyond FAR are statistically significant. Missing baselines: larger models (Qwen-32B, DeepSeek-R1-67B), properly tuned Mol-Instructions.

**Limited scope**: Single dataset (ChEBI-20), no generalization testing. Need OOD evaluation on TOMG [5], Ether0 [6], or other molecular benchmarks. Unclear if results transfer beyond this specific dataset.

**Underspecified details**: L175-176 "semantic consistency filtering" never explained (how measured? criteria? retention rate?). L178 "simulate chemist reasoning" without validation - no chemist review of reasoning quality (only final outputs). L166-171 SELFIES claimed superior to SMILES without empirical evidence. Appendix D: context length not specified.

**Questionable evaluation (Section 3.4, Appendix E)**: Figure 3 format scores differ dramatically between tasks without explanation. No correlation shown between GPT-5 scores and objective metrics (Tables 1-2). "Scoring bias adjustment" instructs GPT-5 to "be more lenient" - appears arbitrary, risks inflating scores to fit narrative. Qualitative analysis done by GPT-5 is unconvincing; would prefer truly independent experts in double-blind manner.

**Metric redundancy**: Frag-J, Frag-R, FG-Match measure overlapping concepts (all assess structural fragment similarity). No evidence they capture distinct properties or justification for needing all three vs. fingerprints alone.

### Additional References

 - [1] Adrian Mirza, Nawaf Alampara, Sreekanth Kunchapu, Martinño Ríos-García, Benedict Emoekabu, Aswanth Krishnan, Tanya Gupta, Mara Schilling-Wilhelmi, Macjonathan Okereke, Anagha Aneesh, et al. Are large language models superhuman chemists? arXiv preprint arXiv:2404.01475, 2024.

 - [2] Yuqing Huang, Rongyang Zhang, Xuesong He, Xuyang Zhi, Hao Wang, Xin Li, Feiyang Xu, Deguang Liu, Huadong Liang, Yi Li, et al. Chemeval: a comprehensive multi-level chemical evaluation for large language models. arXiv preprint arXiv:2409.13989, 2024.

 - [3] Nicholas T. Runcie, Charlotte M. Deane, and Fergus Imrie. Assessing the chemical intelligence of large language models. arXiv:2505.07735, 2025.

 - [4] David Rein, Betty Li Hou, Asa Cooper Stickland, Jackson Petty, Richard Yuanzhe Pang, Julien Dirani, Julian Michael, and Samuel R. Bowman. Gpqa: A graduate-level google-proof q&a benchmark. First Conference on Language Modeling, 2024.

 - [5] Jiatong Li, Junxian Li, Yunqing Liu, Dongzhan Zhou, and Qing Li. Tomg-bench: Evaluating llms on text-based open molecule generation. arXiv preprint arXiv:2412.14642, 2024.

 - [6] Siddharth M Narayanan, James D Braza, Ryan-Rhys Griffiths, Albert Bou, Geemi Wellawatte, Mayk Caldas Ramos, Ludovico Mitchener, Samuel G Rodriques, and Andrew D White. Training a scientific reasoning model for chemistry. arXiv preprint arXiv:2506.17238, 2025.

 - [7] Yanda Chen, Joe Benton, Ansh Radhakrishnan, Jonathan Uesato, Carson Denison, John Schulman, Arushi Somani, Peter Hase, Misha Wagner, Fabien Roger, et al. Reasoning models don't always say what they think. arXiv preprint arXiv:2505.05410, 2025.

 - [8] Jack Lindsey, Wes Gurnee, Emmanuel Ameisen, Brian Chen, Adam Pearce, Nicholas L. Turner, Craig Citro, David Abrahams, Shan Carter, Basil Hosmer, Jonathan Marcus, Michael Sklar, Adly Templeton, Trenton Bricken, Callum McDougall, Hoagy Cunningham, Thomas Henighan, Adam Jermyn, Andy Jones, Andrew Persic, Zhenyi Qi, T. Ben Thompson, Sam Zimmerman, Kelley Rivoire, Thomas Conerly, Chris Olah, and Joshua Batson. On the biology of a large language model. Transformer Circuits Thread, 2025. URL https://transformer-circuits.pub/2025/attribution-graphs/biology.html.

 - [9] Bowen Baker, Joost Huizinga, Leo Gao, Zehao Dou, Melody Y Guan, Aleksander Madry, Wojciech Zaremba, Jakub Pachocki, and David Farhi. Monitoring reasoning models for misbehavior and the risks of promoting obfuscation. arXiv preprint arXiv:2503.11926, 2025.

**Questions:**

**Core Claims & Motivation:**
1. How do you reconcile claims about LLM failures (L46-51) with performance on ChemBench, ChemLLMBench, ChemIQ, and GPQA chemistry subset?
2. How do you address CoT unfaithfulness findings in reference to your claim that your approach delivers interpretable CoT [7]?
3. Why is GRPO appropriate for captioning without verifiable ground truth?

**Methodology:**
4. Precise methodology for "semantic consistency filtering" (L175-176)? Filtering criteria? Retention rate?
5. How was GPT-4o reasoning validated? Did chemists review reasoning quality (not just outputs) per L178?
6. Why assert SELFIES superiority (L166-171) without empirical comparison to SMILES?
7. Why "scoring bias adjustment" in Appendix E? How does this avoid narrative-pushing?

**Statistical Rigor:**
8. Can you provide error bars, confidence intervals, p-values for all metrics? Is 0.0740→0.0758 exact match (Table 4) significant?
9. What's the correlation between GPT-5 scores and objective metrics (Tables 1-2)? Why do format scores differ dramatically in Figure 3?

**Ablations:**
10. Critical missing ablations:
    - Base + RL (same compute) vs. Mol-SFT + Mol-RL?
    - SFT on same data ± CoT annotations plus also RL?
    - Statistical significance of rewards beyond FAR?
11. Were baselines (Mol-Instructions) properly hyperparameter-tuned?

**Reward Design:**
12. Correlation matrix for FPsim, SELFIESsim, FRAGsim, FGmatch? Are they distinct or correlated?
13. Do Frag-J, Frag-R, FG-Match provide information beyond fingerprints? Are improvements on these predictive of downstream performance?

**Generalization:**
14. Evaluation on additional datasets (TOMG[5], others) beyond ChEBI-20?
15. What is the context length (Appendix D)?

---

### Official Review · Reviewer_NwAg · 2025-10-29

**Soundness:** 3
**Presentation:** 3
**Contribution:** 2
**Rating:** 4
**Confidence:** 5

**Summary:**

This paper proposes MolReasoner, a two-stage framework for enhancing molecular reasoning in Large Language Models, enabling interpretable chemical reasoning. Firstly, Mol-SFT trains the model with reasoning trajectories distilled from GPT-4o. Then, Mol-RL utilizes verifiable and multi-level rewards for fine-grained semantic and structural alignment.

**Strengths:**

1. It is practically meaningful to explore molecular reasoning in LLMs.
2. The authors propose novel metrics (e.g., Frag-J, FG-Match), which can capture structural hallucinations beyond validity.
3. The performance is much better than the selected baselines.

**Weaknesses:**

1. The framework is not novel enough. It seems an application of the R1 framework in molecular reasoning.
2. The proposed multi-level molecular rewards actually reward any molecules, which may result in poor performance of generation.
3. Poor baselines. More reasoning LLMs could be included. For example, QWQ and the original version of DeepSeek-R1 should be compared.
4. The ablation study is not convincing. The authors could discuss more about the benefits of the RL stage.
5. The entire framework lacks theoretical support.

**Questions:**

1. Could the authors justify the use of SELFIES in the given context?
2. Please see in Weaknesses.

---

### Official Review · Reviewer_66vY · 2025-10-31

**Soundness:** 3
**Presentation:** 3
**Contribution:** 3
**Rating:** 4
**Confidence:** 3

**Summary:**

In this paper the authors introduced MolReasoner, a two-stage framework for training LLMs on molecular reasoning tasks.
Mol-SFT, which uses GPT-4o-generated Chain-of-Thought (CoT) trajectories enriched with structural features and functional groups to initialize reasoning capabilities, and (2) Mol-RL, which employs reinforcement learning (GRPO) with multi-level rewards to refine reasoning.
Results show MolReasoner achieves 2-4× improvements across metrics including BLEU, ROUGE, fingerprint similarity, validity.

**Strengths:**

The paper addresses a genuine limitation in molecular LLMs—the tendency toward memorization rather than structured reasoning. The distinction between prompt-based methods (lacking domain adaptation), fine-tuning without explicit reasoning (lacking interpretability), and the proposed reasoning-enhanced approach is clearly articulated (Figure 1). The two-stage pipeline (Mol-SFT then Mol-RL) follows a logical progression: warm-up with synthetic CoT data establishes reasoning format, then RL refines chemical accuracy. This design is intuitive and well-justified, avoiding the cold-start problem that would plague pure RL approaches.

The experimental results are impressive: 2.62× improvement in BLEU-2 for captioning, 2.57× in BLEU for generation, 1.43× in MACCS fingerprint similarity, and 1.80× in fragment recall over the best baseline Mol-Instructions + LLaMA3.

**Weaknesses:**

The paper's core contribution is applying existing techniques (CoT distillation + RLHF/GRPO) to molecular tasks, this is incremental engineering on a downstream task rather than methodological innovation. CoT distillation from teacher models (GPT-4o) has been extensively explored in reasoning literature, and GRPO is an established RL algorithm.

The comparison with Mol-Instructions (Table 1) is misleading because Mol-Instructions uses final-answer-only supervision, while MolReasoner uses 42,000 CoT-augmented examples which are fundamentally different training regimes.

**Questions:**

Despite claims of "interpretable reasoning," the paper provides minimal qualitative analysis. Figure 1(c) shows one successful example, but where are the failure cases? What types of reasoning errors persist? Do models hallucinate intermediate steps while arriving at correct final answers?

The paper should include reasoning chain analysis: 1. Typology of reasoning errors e.g. wrong structural decomposition, incorrect functional group identification, logical inconsistencies. 2. Correlation between reasoning quality and final answer correctness—can the model produce correct answers with flawed reasoning or vice versa?

---

### Official Review · Reviewer_9Hrn · 2025-11-01

**Soundness:** 2
**Presentation:** 3
**Contribution:** 2
**Rating:** 2
**Confidence:** 5

**Summary:**

This paper presents MolReasoner, a two-stage training framework with SFT and RL for molecules. The SFT step leverages the GPT-4o generated reasoning enriched with structural information and RL step is trained with the verifiable multi-level rewards.

**Strengths:**

- The authors proposes simple and intuitive framework for molecular reasoning.
- The figure effectively increases the understandability of the paper.

**Weaknesses:**

- **Wrong prior work**: The authors’ discussion of prior work [2] (line 157) appears inaccurate. Although the paper title seems relevant to *structural information reasoning*, the cited work actually focuses on *structured reasoning for chemical equations*, not molecular structure-based reasoning. In addition, a minor but important issue is that the authors list **MSR [1]** and another **Jang et al.** paper (also mentioned at line 157) as distinct works, whereas they are in fact the *same paper* (as evident from the shared arXiv link). The authors should review and revise their citations more carefully to ensure accuracy and avoid redundancy.
- L**ack of clear novelty**: While the authors note that their approach is inspired by prior work [1], the unique contribution of the proposed method remains unclear. Is the novelty primarily derived from the reinforcement learning setup, or from the adoption of SELFIES and GPT-4o where prior works used different settings? If so, how does this distinguish the method from **ether0** [3], an RL-based reasoning framework for chemistry, and **MolLlama** [4], a SFT model leveraging GPT-4o-extracted reasoning? The authors should explicitly articulate the methodological or conceptual advances beyond these existing works.
- **Baseline implementation and fairness of comparison**: The description of **MSR [1]** in lines 317–320 is misleading. MSR does *not* rely on prompting for either task, and even in the molecule captioning setup, the corresponding structural reasoning can be obtained through external tools without explicit LLM prompting. How exactly did the authors implement MSR in their experiments? If they merely prompted the LLM to reason about structures, this would constitute an unfair comparison, since the proposed **MolReasoner** is fine-tuned while MSR is not. Given that both works share almost identical motivations—structure-aware molecular reasoning—an *apple-to-apple* comparison is essential to establish the validity and claimed superiority of the proposed method.


[1] Jang et al., Structural Reasoning Improves Molecular Understanding of LLM, ACL 2025

[2] Ouyang et al., Structured chemistry reasoning with large language models, Arxiv 2023.

[3] Narayanan, et al. Training a Scientific Reasoning Model for Chemistry. NeurIPS 2025.

[4] Kim et al., Mol-LLaMA: Towards General Understanding of
Molecules in Large Molecular Language Model. NeurIPS 2025.

**Questions:**

Reward choice: Why did the authors used the mean of six metrics for R_{gen}? As METEOR works as the F1-score of ROUGE and BLEU, it seems that using solely METEOR is sufficient. I see the ablation study in Table 3 but isn’t it fair to compare between MolReasoner+other metrics(BLEU, METEOR, etc.), not Warm-up+other metrics?

---

### Note · Authors · 2025-11-27

I have read and agree with the venue's withdrawal policy on behalf of myself and my co-authors.